# Return to Tradition: Learning Reliable Heuristics with Classical Machine Learning

## Abstract

There has been a renewed interest in applying machine learning to planning due to recent developments in deep neural networks, with a lot of focus being placed on learning domain-dependent heuristics. However, current approaches for learning heuristics have yet to achieve competitive performance against domain-independent heuristics in several domains, and have poor overall performance. In this work, we construct novel graph representations of lifted planning tasks and use the WL algorithm to generate features from them. These features are used with classical machine learning methods such as Support Vector Machines and Gaussian Processes, which are both fast to train and evaluate. Our novel approach, WL-GOOSE, reliably learns heuristics from scratch and outperforms the $h^{\text{FF}}$ heuristic. It also outperforms or ties with LAMA on 4 out of 10 domains. To our knowledge, the WL-GOOSE learned heuristics are the first to achieve these feats. Furthermore, we study the connections between our novel feature generation methods, previous theoretically flavoured learning architectures, and Description Logic features.

## 1 Introduction

Learning for planning has regained traction in recent years due to advancements in deep learning (DL) and neural network architectures. The focus of learning for planning is to learn domain knowledge in an automated, domain-independent fashion in order to improve the computation and/or quality of plans. Recent examples of learning for planning methods using DL include learning policies (Toyer et al. 2018; Groshev et al. 2018; Garg, Bajpai, and Mausam 2019; Rivlin, Hazan, and Karpas 2020; Ståhlberg, Bonet, and Geffner 2022, 2023), heuristics (Shen, Trevizan, and Thiébaux 2020; Karia and Srivastava 2021) and heuristic proxies (Shen et al. 2019; Ferber et al. 2022). However, learning for planning is not a new field and works capable of learning similar domain knowledge using classical statistical machine learning (SML) methods predate DL. For instance, learning heuristic proxies using support vector machines (SVMs) (Garrett, Kaelbling, and Lozano-Pérez 2016) and learning policies using reinforcement learning (Buffet and Aberdeen 2009) and decision lists (Yoon, Fern, and Givan 2002). We refer to (Jiménez et al. 2012) for a more comprehensive overview of classical SML methods for learning.

A notable advantage of DL methods over current SML methods for learning for planning is their ability to generalise to unseen problems. For instance, DL-based generalised policies have demonstrated scalability to larger problems than previously possible (Toyer et al. 2020), and heuristics learned using GNNs have shown generalisation across different domains (Shen, Trevizan, and Thiébaux 2020). This added generalisation capability of DL comes with certain trade-offs, including a higher demand for data, a large number of hyperparameters and slower training and evaluation times compared to SML methods.

In this paper, we introduce WL-GOOSE, a novel approach for learning for planning that combines the efficiency of SML-based methods with the generalisation capabilities of DL-based methods. WL-GOOSE uses a new graph representation for lifted planning tasks. However, differently from several DL-based methods, we do not use GNNs to learn domain knowledge and use *graph kernels* instead. More precisely, we use a modified version of the Weisfeiler-Lehman algorithm for generating features for graphs (Shervashidze et al. 2011). This allows us to construct kernels using our graphs as input in order to train SML models. Another benefit of WL-GOOSE is its support for various learning targets, such as heuristic values or policies, without the need for backpropagation to generate features as in DL-based approaches. We also provide a theoretical comparison between our approach, GNNs for learning planning heuristics, and Description Logic features (Martín and Geffner 2000).

To demonstrate the potential of WL-GOOSE, we applied it to learn domain-specific heuristics using two classical SML methods: SVMs and Gaussian Processes (GPs). We evaluated the learned heuristics against the state-of-the-art learning for planning on the 2023 International Planning Competition Learning Track benchmarks (Seipp and Segovia-Aguas 2023). The learned heuristics generalised better than previous DL-based methods while also being more computationally efficient: our models took less than 10 seconds to train and are up to 421 times faster than GNNs which train on GPUs. Furthermore, our models were trained in a deterministic fashion with minimal parameter tuning, unlike DL-based approaches which require stochastic gradient descent and tuning of various hyperparameters. When used with greedy best-first search, our learned heuristics models achieved higher total coverage than $h^{\text{FF}}$ (Hoffmann and Nebel 2001) with an increase of 11% and 19% for SVMs and GPs, respectively. Moreover, our learned SVM and GP

models outperformed or tied with LAMA (Richter and Westphal 2010) on 4 out of 10 domains. To our knowledge, these results make our learned heuristics using WL-GOOSE the first ones to surpass the performance of $h^{\text{FF}}$ and the best performing learned heuristics against LAMA.

## 2 Background and Notation

### Planning

A classical planning task (Geffner and Bonet 2013) is a state transition model given by a tuple $\Pi = \langle S, A, s_0, G \rangle$ where $S$ is a set of states, $A$ a set of actions, $s_0 \in S$ an initial state and $G \subseteq S$ a set of goal states. An action $a \in A$ is a function $a : S \to S \cup \perp$ where $a(s) = \perp$ indicates that action $a$ is not applicable at state $s$, and otherwise, $a(s)$ is the successor state when $a$ is applied to $s$. An action also has an associated cost $c(a) \in \mathbb{R}$. A solution or *plan* for this model is a sequence of actions $\pi = a_1 \cdot \ldots \cdot a_n$ where $s_i = a_i(s_{i-1}) \neq \perp$ for $i = 1, \ldots, n$ and $s_n \in G$. In other words, a plan is a sequence of applicable actions which progresses the initial state to a goal state when executed. The cost of a plan $\pi$ is the sum of its action costs: $c(\pi) = \sum_{i=1}^{n} c(a_i)$. A planning task is *solvable* if there exists at least one plan. A plan is *optimal* if there does not exist any other plan with strictly lower cost.

We represent planning tasks in a compact form which does not require enumerating all states and actions. A *lifted planning task* (Lauer et al. 2021) is a tuple $\Pi = \langle \mathcal{P}, \mathcal{O}, \mathcal{A}, s_0, G \rangle$ where $\mathcal{P}$ is a set of first-order predicates, $\mathcal{O}$ a set of objects, $\mathcal{A}$ a set of action schemas, $s_0$ the initial state, and $G$ the goal condition. A predicate $P \in \mathcal{P}$ has a set of parameters $x_1, \ldots, x_{n_P}$ where $n_P \in \mathbb{N}$ depends on $P$, and it is possible for a predicate to have no parameters. A ground proposition is a predicate which is instantiated by assigning all of the $x_i$ with objects from $\mathcal{O}$ or other defined variables. An action schema $a \in \mathcal{A}$ is a tuple $\langle \Delta(a), \text{pre}(a), \text{add}(a), \text{del}(a) \rangle$ where $\Delta(a)$ is a set of parameter variables, and the preconditions $\text{pre}(a)$, add effects $\text{add}(a)$, and delete effects $\text{del}(a)$ are sets of predicates from $\mathcal{P}$ instantiated with elements from $\Delta(a) \cup \mathcal{O}$. An action is an action schema where each variable is instantiated with an object. A domain $\mathcal{D}$ is a set of lifted planning tasks which share the same sets of predicates $\mathcal{P}$ and action schemas $\mathcal{A}$.

In a lifted planning task, states are represented as sets of ground propositions. The following are sets of ground propositions: states, goal condition, and the preconditions, add effects, and delete effects of all actions. An action $a$ is applicable in a state $s$ if $\text{pre}(a) \subseteq s$, in which case we define $a(s) = (s \setminus \text{del}(a)) \cup \text{add}(a)$. Otherwise $a(s) = \perp$. The cost of an action is given by the cost of its corresponding action schema. A state $s$ is a goal state if $G \subseteq s$.

A heuristic is a function $h : S \to \mathbb{R} \cup \{\infty\}$ which maps a state into a real number representing an estimate of the cost of the optimal plan to the goal, or $\infty$ representing that the state is unsolvable. A heuristic can be defined on problems by evaluating the heuristic at their initial state: $h(\Pi) = h(s_0)$. The optimal heuristic $h^*$ returns for each state $s$ the cost of the optimal plan to the goal if the problem is solvable from $s$, and $\infty$ otherwise.

---

**Algorithm 1:** WL algorithm
1   $c^0(v) \leftarrow c(v), \forall v \in V$
2   **for** $j = 1, \ldots, h$ **do for** $v \in V$ **do**
3    $c^j(v) \leftarrow \text{hash}\left(c^{j-1}(v), \{\!\!\{c^{j-1}(u) \mid u \in \mathcal{N}(v)\}\!\!\}\right)$
4   **return** $\bigcup_{j=0,\ldots,h}\{\!\!\{c^j(v) \mid v \in V\}\!\!\}$

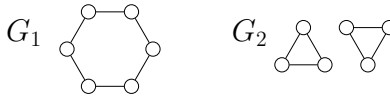

Figure 1: Two non-isomorphic graphs $G_1$ (6-cycle) and $G_2$ (two disjoint 3-cycles) which the WL algorithm returns the same outputs, thus failing to distinguishing them.

### The Weisfeiler-Lehman algorithms

We write $\langle V, E, c, l \rangle$ for a graph with coloured nodes and edges, where $V$ is a set of nodes, $E \subseteq \binom{V}{2}$ is a set of undirected edges, $c : V \to \Sigma_V$ maps nodes to a set of colours $\Sigma_V$, and $l : E \to \Sigma_E$ maps edges to a set of colours $\Sigma_E$. The edge neighbourhood of a node $u$ under edge colour $\iota$ is $\mathcal{N}_\iota(u) = \{e = \langle u, v \rangle = \langle v, u \rangle \in E \mid l(e) = \iota\}$. The neighbourhood of a node $u$ in a graph is $\mathcal{N}(u) = \bigcup_{\iota \in \Sigma_E} \mathcal{N}_\iota(u)$.

We only focus on the WL algorithm which is a special case of the class of $k$-Weisfeiler-Lehman ($k$-WL) algorithms (Leman and Weisfeiler 1968). The $k$-WL algorithms were originally constructed to provide tests for whether pairs of graphs are isomorphic or not. The $k+1$-WL algorithm subsumes the $k$-WL algorithm as it can distinguish a greater class of non-isomorphic graphs, and furthermore is in correspondence with $k$-variable counting logics (Cai, Fürer, and Immerman 1992). However, the complexity of the $k$-WL algorithms is exponential in $k$.

The WL algorithm takes as input graphs without edge colours, i.e. $\forall e \in E, l(e) = 0$, and outputs a canonical form in terms of a multiset of colours, a set which is allowed to have duplicate elements. It has also been used to construct a kernel for graphs (Shervashidze et al. 2011) which converts the multiset of colours in the WL algorithm into a feature vector and then uses the simple dot product kernel. We denote a multiset of elements by $\{\!\!\{ \ldots \}\!\!\}$.

The WL algorithm is shown in Alg. 1 which takes as input a graph $G$ with coloured nodes only and a predefined number of WL iterations $h$. The algorithm begins by initialising the current colours of each node with the initial node colours. If no node colours are given in the graph, we can set them to 0. Line 3 updates the colour of each node $v$ by iteratively collecting the current colours of its neighbors in a multiset and then hashing this multiset and $v$'s current colour into a colour using an injective hash$(\cdot, \cdot)$ function. In practice, hash is built lazily by using a map data structure and multisets are represented as sorted strings. Line 4 returns a multiset of the node colours seen over all iterations.

If the WL algorithm outputs two different multisets for two graphs $G_1$ and $G_2$, then the graphs are non-isomorphic. However, if the algorithm outputs the same multisets for two graphs we cannot say for sure whether they are isomor-

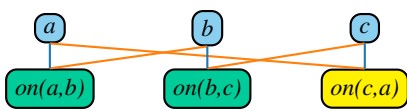

Figure 2: ILG subgraph of facts and goal condition corresponding to the *on* predicate of a Blocksworld instance. The current state says that $a$ stacked on $b$, which is stacked on $c$, and the goal condition is for $c$ to be stacked on $a$.

phic or not. The canonical example illustrating this case is in Fig. 1 where the two graphs are not isomorphic but the WL algorithm returns the same output for both graphs since it views all nodes as the same because they have degree 2.

## 3    WL Features for Planning

In this section we describe how to generate features for planning states in order to learn heuristics. The process involves three main steps: (1) converting planning states into graphs with coloured nodes and edges, (2) running a variant of the WL algorithm on the graphs in order to generate features, and then (3) training a classical machine learning model for predicting heuristics using the obtained features. We start by defining the Instance Learning Graph (ILG), a novel representation for lifted planning tasks.

**Definition 3.1.** The *instance learning graph (ILG)* of a lifted planning problem $\Pi = \langle \mathcal{P}, \mathcal{O}, \mathcal{A}, s_0, G \rangle$ is the graph $G = \langle V, E, c, l \rangle$ with

- $V = \mathcal{O} \cup s_0 \cup G$
- $E = \bigcup_{p=P(o_1,\ldots,o_{n_P}) \in s_0 \cup G} \{\langle p, o_1 \rangle, \ldots, \langle p, o_{n_P} \rangle\}$
- $c : V \to (\{\texttt{ap}, \texttt{ug}, \texttt{ag}\} \times \mathcal{P}) \cup \{\texttt{ob}\}$ defined by

$$
u \mapsto \begin{cases}
\texttt{ob}, & \text{if } u \in \mathcal{O}; \\
(\texttt{ag}, P), & \text{if } u = P(o_1, \ldots, o_{n_P}) \in s_0 \cap G; \\
(\texttt{ap}, P), & \text{if } u = P(o_1, \ldots, o_{n_P}) \in s_0 \setminus G; \\
(\texttt{ug}, P), & \text{if } u = P(o_1, \ldots, o_{n_P}) \in G \setminus s_0;
\end{cases}
$$

- $l : E \to \mathbb{N}$ with $\langle p, o_i \rangle \mapsto i$.

Fig. 2 provides an example of an ILG. An ILG consists of a node for each object and the union of propositions that are true in the state $s_0$ and the goal condition $G$. A proposition is connected to the $n$ object nodes which instantiates the proposition. The labels of the $n$ edges correspond to the position of the object in the predicate argument. The colours of the nodes indicate whether the node corresponds to an object (ob), or determines whether it is a proposition belonging to $s_0$ (ap) or $G$ (ug) only or both (ag), as well as its corresponding predicate. Hence ug stands for unachieved goal, ag for achieved goal, and ap for achieved (non-goal) proposition. Note that ILGs are agnostic to the transition system of the planning task as they ignore action schemas and actions.

Since ILGs have coloured edges, we need to extend the WL algorithm to account for edge colours to generate features for ILGs. Our modified WL algorithm is obtained by replacing Line 3 in Alg. 1 with the update function

$$
c^j(v) \leftarrow f\Big(c^{j-1}(v), \bigcup_{\iota \in \Sigma_E} \big\{\!\!\big\{ (c^{j-1}(u), \iota) \mid u \in \mathcal{N}_\iota(v) \big\}\!\!\big\}\Big),
$$

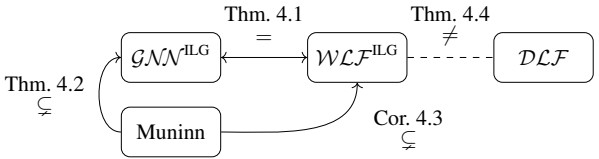

Figure 3: Expressivity hierarchy of WL, GNN and DL generated features for planning.

where the union of multisets is itself a multiset. Note that edge colours do not update during this modified WL algorithm. It is possible to run a variant of the WL algorithm which modifies edge colours but this comes at an additional computational cost given that usually $|E| \gg |V|$.

Now that ILGs can be represented as multisets of colours, we can generate features by representing these multisets as histograms (Shervashidze et al. 2011). The feature vector of a graph is a vector $v$ with a size equal to the number of observed colours during training, where $v[\kappa]$ counts how many times the WL algorithm has encountered colour $\kappa$ throughout its iterations. Formally, let $G_1 = \langle V_1, E_1, c_1, l_1 \rangle, \ldots, G_n = \langle V_n, E_n, c_n, l_n \rangle$ be the set of training graphs. Then the colours the WL algorithm encounters in the training graphs are given by

$$
\mathcal{C} = \{c_i^j(v) \mid i \in \{1, \ldots, n\}; j \in \{0, \ldots, h\}; v \in V_i\}
$$

where $c_i^j(v)$ is the colour of node $v$ in graph $G_i$ during the $j$-th iteration of WL for $j > 0$ and $c_i^0(v) = c_i(v)$. Given a planning task $\Pi$ and the set of colours $\mathcal{C}$ observed during training, $\Pi$'s feature vector representation $\vec{v} \in \mathbb{R}^{|\mathcal{C}|}$ is $\vec{v} = [\texttt{count}_\mathcal{C}(\Pi, \kappa_1), \ldots, \texttt{count}_\mathcal{C}(\Pi, \kappa_{|\mathcal{C}|})]$ where $\texttt{count}_\mathcal{C}(\Pi, \kappa)$ is the number of times the colour $\kappa \in \mathcal{C}$ is present in the output of the WL algorithm on the ILG representation of $\Pi$. There is no guarantee that $\mathcal{C}$ contains all possible observable colours for a given planning domain and colours not in $\mathcal{C}$ observed after training are ignored.

## 4    Theoretical Results

In this section, we investigate the relationship between WL features and features generated using message passing graph neural network (GNN), description logic features (DLF) for planning (Martín and Geffner 2000) and the features used by Muninn (Ståhlberg, Bonet, and Geffner 2022, 2023), a theoretically motivated deep learning model. Fig. 3 summarises our theoretical results.

We begin with some notation. Let $\mathcal{D}$ represent the set of all problems in a given domain. We define $\mathcal{WLF}_\Theta^{\text{ILG}} : \mathcal{D} \to \mathbb{R}^d$ as the WL feature generation function described in Sec. 3 which runs the WL algorithm on the ILG representation of planning tasks. We denote $\Theta$ the set of parameters of the function which includes the number of WL iterations and the set of colours $\mathcal{C}$ with size $d$ observed during training. We similarly denote parametrised GNNs acting on ILG representations of planning tasks by $\mathcal{GNN}_\Theta^{\text{ILG}} : \mathcal{D} \to \mathbb{R}^d$. Parameters for GNNs include number of message passing layers, the message passing update function with fixed weights, and the aggregation function.

We denote DLF generators (Martín and Geffner 2000) by $\mathcal{DLF}_{\Theta} : \mathcal{D} \to \mathbb{R}^d$ where the parameters for $\mathcal{DLF}$ include the maximum complexity length of its features (Bonet, Francès, and Geffner 2019). DLFs have been used in several areas of learning for planning including learning descending dead-end avoiding heuristics (Francès et al. 2019), unsolvability heuristics (Ståhlberg, Francès, and Seipp 2021) and policy sketches (Drexler, Seipp, and Geffner 2022). Lastly, we denote the architecture from Ståhlberg, Bonet, and Geffner (2022) for generating features by Muninn$_{\Theta}$ : $\mathcal{D} \to \mathbb{R}^d$. We omit their final MLP layer which transforms the vector feature into a heuristic estimate. Furthermore in our theorems, we ignore their use of random node initialisation (RNI) (Abboud et al. 2021). The original intent of RNI is to provide a universal approximation theorem for GNNs but the practical use of the theorem is limited by the assumption of exponential width layers and absence of generalisation results. Thus, it is unclear what benefits RNI brings to Muninn for planning. Parameters for Muninn include hyperparameters for their GNN architecture and the learned weights for their update functions.

In all of the aforementioned models, the parameters $\Theta$ consist of a combination of model hyperparameters and trained parameters based on a training set $\mathcal{T}_{\mathcal{D}} \subseteq \mathcal{D}$. The expressivity and distinguishing power of a feature generator for planning determines if it can theoretically learn $h^*$ for larger subsets of planning tasks. We begin with an application of a well-known result connecting the expressivity of the WL algorithm and GNNs for distinguishing graphs (Xu et al. 2019) by extending it to edge-labelled graphs.

**Theorem 4.1** ($\mathcal{WLF}^{\text{ILG}}$ and $\mathcal{GNN}^{\text{ILG}}$ have the same power at distinguishing planning tasks.). *Let* $\Pi_1$ *and* $\Pi_2$ *be any two planning tasks from a given domain. If for a set of parameters* $\Theta$ *we have that* $\mathcal{GNN}^{\text{ILG}}_{\Theta}(\Pi_1) \neq \mathcal{GNN}^{\text{ILG}}_{\Theta}(\Pi_2)$, *then there exists a corresponding set of parameters* $\Phi$ *such that* $\mathcal{WLF}^{\text{ILG}}_{\Phi}(\Pi_1) \neq \mathcal{WLF}^{\text{ILG}}_{\Phi}(\Pi_2)$. *Conversely for all* $\Phi$ *such that* $\mathcal{WLF}^{\text{ILG}}_{\Phi}(\Pi_1) \neq \mathcal{WLF}^{\text{ILG}}_{\Phi}(\Pi_2)$, *there exists* $\Theta$ *such that* $\mathcal{GNN}^{\text{ILG}}_{\Theta}(\Pi_1) \neq \mathcal{GNN}^{\text{ILG}}_{\Theta}(\Pi_2)$.

*Proof.* [$\subseteq$] The forward statement follows from (Xu et al. 2019, Lemma 3) which states that GNNs are at most as expressive as the WL algorithm for distinguishing non-isomorphic graphs. We can modify the lemma for the edge labelled WL algorithm and GNNs which account for edge features. Then the result follows after performing the transformation of planning tasks into the ILG representation.

[$\supseteq$] The converse statement follows from (Xu et al. 2019, Corollary 6) and modifying Eq. (4.1) of their GIN architecture by introducing an MLP for each of the finite number of edge labels in the ILG graph and summing their outputs at each GIN layer. The MLPs have disjoint range in order for injectivity to be preserved as to achieve the same distinguishing power of the edge labelled WL algorithm. This can be easily enforced by increasing the hidden dimension size and having each MLP to map to orthogonal dimensions. $\square$

We proceed to show that GNNs acting on ILGs is similar to Muninn's GNN architecture (Ståhlberg, Bonet, and

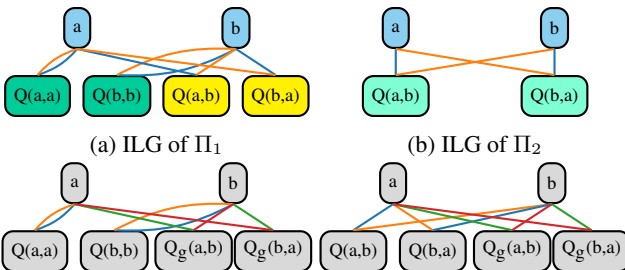

(a) ILG of $\Pi_1$     (b) ILG of $\Pi_2$

(c) Implicit Muninn graph of $\Pi_1$ (d) Implicit Muninn graph of $\Pi_2$

Figure 4: ILG and Muninn graph representations of tasks in Thm. 4.2 [$\supsetneq$].

Geffner 2022). The idea of the proof is that encoding different predicates into the ILG representation is equivalent to having different weights for message passing to and from different predicates in Muninn. However, we also show that our model has strictly higher expressivity for distinguish planning tasks due to explicitly encoding achieved goals.

**Theorem 4.2** ($\mathcal{GNN}^{\text{ILG}}$ is strictly more expressive than Muninn at distinguishing planning tasks.). *Let* $\Pi_1$ *and* $\Pi_2$ *be any two planning tasks from a given domain. For all* $\Theta$, *if* $Muninn_{\Theta}(\Pi_1) \neq Muninn_{\Theta}(\Pi_2)$, *then there exists a corresponding set of parameters* $\Phi$ *such that* $\mathcal{GNN}^{\text{ILG}}_{\Phi}(\Pi_1) \neq \mathcal{GNN}^{\text{ILG}}_{\Phi}(\Pi_2)$. *Furthermore, there exists a pair of planning tasks* $\Pi_1$ *and* $\Pi_2$ *such that there exists* $\Phi$ *with* $\mathcal{GNN}^{\text{ILG}}_{\Phi}(\Pi_1) \neq \mathcal{GNN}^{\text{ILG}}_{\Phi}(\Pi_2)$ *but for all* $\Theta$, *$Muninn_{\Theta}(\Pi_1) = Muninn_{\Theta}(\Pi_2)$*.

*Proof.* [$\supseteq$] In order to show the inclusion, we show that a Muninn instance operating on a state can be expressed as a GNN operating on the ILG representation of the state. More explicitly, we show that the implicit graph representation of planning states by Muninn is the same graph as ILG. The message passing steps and initial node features are different but the semantic meaning of executing both algorithms are the same. The node features in the implicit graphs of Muninn are all the same when ignoring random node initialisation. Muninn differentiates object nodes and fact nodes by using different message passing functions depending on whether a node is an object or a fact, and depending on which predicate the fact belongs to. In the language of ILG, Muninn's message passing step on fact nodes $p = P(o_1, \ldots, o_{n_P})$ is

$$h_p^{l+1} = \textbf{MLP}_P(h_{o_1}^l, \ldots, h_{o_{n_P}}^l) \quad (1)$$

where $h_p^{l+1}$ denotes the latent embedding of the node $p$ in the $l + 1$-th layer, $h_{o_i}^l$ denotes the latent embedding of the object node $o_i$ in the $l$-th layer, and $\textbf{MLP}_P$ is a multilayer perceptron, with a different one for each predicate. The message passing step of Muninn on object nodes $o_i$ is

$$h_o^{l+1} = \textbf{MLP}_U(h_o^l, \{\!\{h_p^l \mid o \in p\}\!\}) \quad (2)$$

where $o \in p$ denotes that $o$ is an argument of the predicate associated with $p$. We note that having a different **MLP** in the message passing step for different nodes is equivalent

to having a larger but identical **MLP** in the message passing step for all nodes. This is because the model can learn to partition latent node features depending on their semantic meaning and thus be able to use a single **MLP** function to act as multiple different functions for different node feature partitions. Thus, Eq. (1) and (2) can be imitated by a GNN operating on ILG since ILG features differentiate nodes depending on whether they correspond to an object, or a fact associated with a predicate. Different edge labels in the ILG allow it to distinguish the relationship between facts and objects depending on their position in the predicate argument.

[$\supseteq$] To see how $\mathcal{GNN}^{\text{ILG}}$ is strictly more expressive than Muninn, we consider the following pair of planning tasks. The main idea is that Muninn does not keep track of achieved goals and sometimes cannot even see that the goal has been achieved. Let $\Pi_1 = \langle \mathcal{P}, \mathcal{O}, \mathcal{A}, s_0^{(1)}, G \rangle$ and $\Pi_2 = \langle \mathcal{P}, \mathcal{O}, \mathcal{A}, s_0^{(2)}, G \rangle$ with $\mathcal{P} = \{Q\}$, $\mathcal{O} = \{a, b\}$, $\mathcal{A} = \emptyset$, $G = s_0^{(2)} = \{Q(a,b), Q(b,a)\}$ and $s_0^{(1)} = \{Q(a,a), Q(b,b)\}$. Fig. 4 illustrates the ILG representation and the implicitly defined edge-labelled graph representation in Muninn's GNN architecture of $\Pi_1$ and $\Pi_2$. It is clear that the ILG representation of $\Pi_1$ and $\Pi_2$ are different and hence $\mathcal{GNN}^{\text{ILG}}$ differentiates between $\Pi_1$ and $\Pi_2$. On the other hand without RNI, Muninn sees the pair of non-isomorphic graphs in Fig. 4(c) and (d). However, any edge-labelled variant of the WL algorithm views the pair of graphs as the same, and hence so does Muninn. $\qquad\square$

**Corollary 4.3** ($\mathcal{WLF}^{\text{ILG}}$ *is strictly more expressive than Muninn at distinguishing planning tasks.*)**.** *Let $\Pi_1$ and $\Pi_2$ be any two planning tasks from a given domain. For all $\Theta$, if $\text{Muninn}_\Theta(\Pi_1) \neq \text{Muninn}_\Theta(\Pi_2)$, then there exists a corresponding set of parameters $\Phi$ such that $\mathcal{WLF}_\Phi^{\text{ILG}}(\Pi_1) \neq \mathcal{WLF}_\Phi^{\text{ILG}}(\Pi_2)$. Furthermore, there exists a pair of planning tasks $\Pi_1$ and $\Pi_2$ such that there exists $\Phi$ with $\mathcal{WLF}_\Phi^{\text{ILG}}(\Pi_1) \neq \mathcal{WLF}_\Phi^{\text{ILG}}(\Pi_2)$ but for all $\Theta$, $\text{Muninn}_\Theta(\Pi_1) = \text{Muninn}_\Theta(\Pi_2)$.*

Our next theorem shows that $\mathcal{WLF}^{\text{ILG}}$ and $\mathcal{DLF}$ features are incomparable, in the sense that there are pairs of planning tasks that look equivalent to one model but not the other. We use a similar counterexample to that used for Muninn but with an extra predicate which $\mathcal{WLF}^{\text{ILG}}$ does not distinguish but $\mathcal{DLF}$ can. Conversely we use the fact that DLFs are limited by the need to convert planning predicates into binary predicates to construct a counterexample pair of planning tasks with ternary predicates which $\mathcal{DLF}$ views as the same while $\mathcal{WLF}^{\text{ILG}}$ does not.

**Theorem 4.4** ($\mathcal{WLF}^{\text{ILG}}$ *and $\mathcal{DLF}$ are incomparable at distinguishing planning tasks.*)**.** *There exists a pair of planning tasks $\Pi_1$ and $\Pi_2$ such that there exists $\Phi$ with $\mathcal{WLF}_\Phi^{\text{ILG}}(\Pi_1) \neq \mathcal{WLF}_\Phi^{\text{ILG}}(\Pi_2)$ but for all $\Theta$, $\mathcal{DLF}_\Theta(\Pi_1) = \mathcal{DLF}_\Theta(\Pi_2)$. Furthermore, there exists a pair of planning tasks $\Pi_1$ and $\Pi_2$ such that there exists $\Phi$ with $\mathcal{DLF}_\Phi(\Pi_1) \neq \mathcal{DLF}_\Phi(\Pi_2)$ but for all $\Theta$, $\mathcal{WLF}_\Theta^{\text{ILG}}(\Pi_1) = \mathcal{WLF}_\Theta^{\text{ILG}}(\Pi_2)$.*

*Proof.* [$\exists<$] We begin by describing a pair of planning

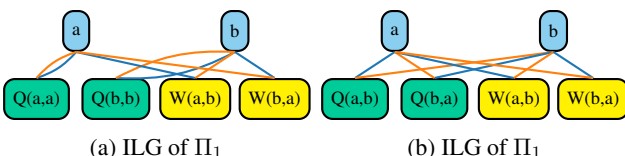

(a) ILG of $\Pi_1$        (b) ILG of $\Pi_1$

Figure 5: ILG representations of tasks in Thm. 4.4 [$\exists<$].

tasks $\Pi_1$ and $\Pi_2$ such that $\mathcal{WLF}_\Theta^{\text{ILG}}(\Pi_1) = \mathcal{WLF}_\Theta^{\text{ILG}}(\Pi_2)$ for any set of parameters $\Theta$ but are distinguished by Description Logics. Let $\Pi_1 = \langle \mathcal{P}, \mathcal{O}, \mathcal{A}, s_0^{(1)}, G \rangle$ and $\Pi_2 = \langle \mathcal{P}, \mathcal{O}, \mathcal{A}, s_0^{(2)}, G \rangle$ with $\mathcal{P} = \{Q, W\}$, $\mathcal{O} = \{a, b\}$, $\mathcal{A}$ contains the single action schema $o = \langle \{x, y\}, \{Q(x,y)\}, \{W(x,y)\}, \emptyset \rangle$, $G = \{W(a,b), W(b,a)\}$, $s_0^{(1)} = \{Q(a,a), Q(b,b)\}$ and $s_0^{(2)} = \{Q(a,b), Q(b,a)\}$.

We have that $h^*(\Pi_1) = \infty$ as the problem $\Pi_1$ is unsolvable, while $h^*(\Pi_2) = 2$ as the optimal plan contains actions $o(a,b)$ and $o(b,a)$. DL features are able to distinguish the two planning tasks by considering the role-value map $(Q = W)(s)$ defined by $\{x \mid \forall y : Q(x,y) \in s \iff W(x,y) \in s\}$, and the corresponding numerical feature $|Q = W|(s) = |(Q = W)(s)|$. We have that $|Q = W|(s_0^{(1)}) = 0$ and $|Q = W|(s_0^{(1)}) = 2$, meaning that $\mathcal{DLF}$ can distinguish between $\Pi_1$ and $\Pi_2$.

On the other hand, the ILG representations of $\Pi_1$ and $\Pi_2$ are indistinguishable to our definition of the edge-labelled WL algorithm. Fig. 5 illustrates this example and we note that it is similar to the implicit Muninn graph representations of the pair of planning tasks from Thm. 4.2.

[$\exists>$] We identify a pair of problems with ternary predicates which compile to the same problem with only binary predicates for which DL features are defined. For problems with at most binary predicates, DL introduces base roles on each predicate $P(x, y) \in \mathcal{P}$ by $P^s = \{(a, b) \mid P(a, b) \in s\}$ where $s$ is a planning state. Then given an $n$-ary predicate $R(x_1, \ldots, x_n)$, DL introduces $n(n-1)/2$ roles defined by $R_{i,j}^s = \{(a, b) \mid \exists o_1, \ldots, o_{i-1}, o_{i+1}, \ldots, o_{j-1}, o_{j+1}, \ldots, o_n \in \mathcal{O}, R(o_1, \ldots, o_{i-1}, a, o_{i+1}, \ldots, o_{j-1}, b, o_{j+1}, \ldots, o_n) \in s\}$ for $1 \le i < j \le n$. Now consider the problems $\Pi_1 = \langle \mathcal{P}, \mathcal{O}, \mathcal{A}, s_0^{(1)}, G \rangle$ and $\Pi_2 = \langle \mathcal{P}, \mathcal{O}, \mathcal{A}, s_0^{(2)}, G \rangle$ now with $\mathcal{P} = \{P\}$, $\mathcal{O} = \{a, b, c, d\}$, $\mathcal{A} = \emptyset$, $G = \{P(a,b,c)\}$, and

$$s_0^{(1)} = \{P(a,b,a), P(c,b,c), P(a,d,c), P(c,d,a)\}$$
$$s_0^{(2)} = \{P(a,b,c), P(c,b,a), P(a,d,a), P(c,d,c)\}.$$

We have that $h^*(\Pi_1) = \infty$ since there are no actions and the initial state is not the goal condition, while $h^*(\Pi_2) = 0$ since $G \subseteq s_0^{(2)}$. The ILG for the two tasks are distinguished by the WL algorithm as the ILG of $\Pi_1$ has no achieved goal colour while $\Pi_2$ does. However, DL features view the two states $s_0^{(1)}$ and $s_0^{(2)}$ as the same due after the compilation from ternary to binary predicates:

| | | | |
|---|---|---|---|
| $P_{1,2}(a,b)$ | $P_{1,2}(a,d)$ | $P_{1,2}(c,b)$ | $P_{1,2}(c,d)$ |
| $P_{1,3}(a,a)$ | $P_{1,3}(a,c)$ | $P_{1,3}(c,a)$ | $P_{1,3}(c,c)$ |
| $P_{2,3}(b,a)$ | $P_{2,3}(b,c)$ | $P_{2,3}(d,a)$ | $P_{2,3}(d,c)$. |

Thus any DL features will be the same for both $s_0^{(1)}$ and $s_0^{(2)}$ and thus cannot distinguish $\Pi_1$ and $\Pi_2$. $\qquad\square$

Our final theorem combines previous results and states that there exist domains for which all feature generators defined thus far are not powerful enough to perfect learn $h^*$. Although this is not a surprising result, we hope to bring some intuition on what is further required for constructing more expressive features for learning for planning.

**Corollary 4.5** (All feature generation models thus far cannot generate features that allow us to learn $h^*$ for all domains.)**.** *Let* $\mathcal{F} \in \left\{ \mathcal{WLF}^{ILG}, \mathcal{GNN}^{ILG}, Muninn, \mathcal{DLF} \right\}$. *There exists a domain* $\mathcal{D}$ *with a pair of planning tasks* $\Pi_1$, $\Pi_2$ *such that for all parameters* $\boldsymbol{\Theta}$ *for* $\mathcal{F}$, *we have that* $\mathcal{F}_{\boldsymbol{\Theta}}(\Pi_1) = \mathcal{F}_{\boldsymbol{\Theta}}(\Pi_2)$ *and* $h^*(\Pi_1) \neq h^*(\Pi_2)$.

In this section, we concluded that our $\mathcal{WLF}^{ILG}$ features are one of the most expressive features thus far in the literature for representing planning tasks, the other being $\mathcal{DLF}$ features. We have done so by drawing an expressivity hierarchy between our $\mathcal{WLF}^{ILG}$ features and previous work on GNN architectures (Ståhlberg, Bonet, and Geffner 2022). We further constructed explicit counterexamples illustrating the difference between $\mathcal{WLF}^{ILG}$ and $\mathcal{DLF}$ features, highlighting their respective advantages and limitations.

# 5 Experiments

In this section, we empirically evaluate WL-GOOSE for learning domain-specific heuristics against the state-of-the-art. We consider the domains and training and test sets from the learning track of the 2023 International Planning Competition (IPC) (Seipp and Segovia-Aguas 2023). The domains are Blocksworld, Childsnack, Ferry, Floortile, Miconic, Rovers, Satellite, Sokoban, Spanner, and Transport. Each domain contains instances categorised into easy, medium and hard difficulties depending on the number of objects in the instance. For each domain, the training set consists of at most 99 easy instances and the test set consists of exactly 30 instances from each of the three easy, medium and hard difficulties that are not in the training set.

The hyperparameters considered for WL-GOOSE are the number of iterations $h$ for generating features using the WL algorithm and the choice of a machine learning model used and its corresponding hyperparameters. In all our experiments with WL-GOOSE, we use $h = 4$ and, since our learning target is $h^*$, we consider the following regression models: support vector regression with the dot product kernel (SVR) and the radial basis kernel (SVR$_\infty$), and Gaussian process regression with the dot product kernel (GPR). We choose SVR over ridge regression for our kernelised linear model due to its sparsity and hence faster evaluation time with use of the $\epsilon$-insensitive loss function (Vapnik 2000). The choice of Gaussian process allows us to explore a Bayesian treatment for learning $h^*$ providing us with confidence bounds on the learned heuristics.

Furthermore, we experiment with the *2-LWL algorithm* with $h = 4$ for generating features alongside SVR with the dot product kernel (2-LWL). The 2-LWL algorithm is a computationally feasible approximation of the 2-WL algorithm (Morris, Kersting, and Mutzel 2017), which in turn is a generalisation of the WL algorithm where colours are assigned to pairs of vertices. While the features computed by the 2-WL algorithm subsume those of the WL algorithm, it is computationally more expensive to compute requiring quadratically more time than the WL algorithm.

For any configuration of WL-GOOSE, we use optimal plans returned by scorpion (Seipp, Keller, and Helmert 2020) on the training set with a 30-minute timeout on each instance for training. We use 67% of the states and the corresponding cost to the goal from each optimal plan as training data and the remaining 33% were used as a validation set for logging loss scores.

As baselines for heuristics, we use the domain-independent heuristic $h^{FF}$ and GNNs operating on the ILG representations of planning tasks. For the GNNs, we use 4 message passing layers and consider both max and mean aggregators. All heuristics are evaluated using greedy best-first search. We include LAMA (Richter and Westphal 2010) using its first plan output as a strong satisficing planner that uses multi-queue heuristic search and other optimisation techniques. We also compare against Muninn using the same weights trained in its entry of the 2023 IPC Learning Track. For all methods, we use a timeout of 1800 seconds per evaluation problem. With the exception of the GNNs using ILGs, the experiments were run on a cluster with single Intel Xeon 3.2 GHz CPU cores and a memory limit of 8GB. For the GNNs using ILGs, we used single Intel Xeon Gold 5218R 2.1GHz CPU cores and an NVIDIA RTX A6000 GPU with 48GB memory. However the GNNs never use more than 8GB of GPU or main memory before reaching the timeout. The other planners in the competition were not considered because they do not explicitly learn a heuristic.

Tab. 1 summarises our experiments by showing the coverage per domain for each planner as well as their total IPC score. Next, we discuss our results in detail and we conclude this section by describing how to analyse the learned features of our models using an example.

**How well do heuristics learned from WL features perform?** Considering total coverage and total IPC score (Tab. 1), we notice that SVR and GPR outperform all the other planners with the exception of LAMA-first, i.e., all learning-based approaches as well as $h^{FF}$. Domain-wise, SVR and GPR outperform Muninn in all domains while outperforming GNNs using ILGs on 8 and 9 domains, respectively. Both SVR and GPR outperform or tie with LAMA on 4 domains, namely Blocksworld, Ferry, Miconic and Spanner. GPR is able to return better plans than LAMA on 5 domains (Blocksworld, Childsnack, Ferry, Miconic, Sokoban), while the reverse is true only on 3 domains (Rovers, Satellite, Spanner). For Spanner, this is because LAMA's heuristics are not informative for this domain, which leads to it performing like blind search and hence returning better plans.

SVR and GPR also outperform or tie with $h^{FF}$ on 5 and 7 domains, respectively. We compare GPR and $h^{FF}$ in more detail in Fig. 6 by showing plan cost and nodes expanded per problem. We observe that the better performing planner on a

| Domain | classical | | GNN | | | WL | | | |
|---|---|---|---|---|---|---|---|---|---|
| | LAMA-F | $h^{FF}$ | Muninn | GNN$_{max}$ | GNN$_{mean}$ | SVR | SVR$_\infty$ | 2-LWL | GPR |
| blocksworld | 61 | 28 | 40 | 49 | 58 | 72 | 20 | 22 | **77** |
| childsnack | 35 | 26 | 11 | 19 | 20 | 16 | 15 | 13 | **30** |
| ferry | 68 | 68 | 46 | 64 | 72 | 77 | 32 | 60 | 76 |
| floortile | 11 | **12** | - | - | - | 2 | - | 1 | 2 |
| miconic | 90 | 90 | 30 | 90 | 90 | 90 | 31 | 67 | 90 |
| rovers | 67 | 34 | 15 | 25 | 29 | 33 | 27 | 33 | **37** |
| satellite | 89 | **65** | 18 | 31 | 29 | 46 | 29 | 20 | 57 |
| sokoban | 40 | 36 | 26 | 32 | 33 | **38** | 30 | 31 | **38** |
| spanner | 30 | 30 | 32 | 30 | 33 | **74** | 30 | 52 | **74** |
| transport | 66 | **41** | 17 | 38 | 35 | 29 | 26 | 35 | 32 |
| sum coverage | 557 | 430 | 235 | 378 | 399 | 477 | 240 | 334 | **513** |
| sum IPC score | 492.7 | 393.5 | 232.4 | 354.5 | 374.6 | 442.8 | 213.4 | 304.8 | **471.2** |

Table 1: Coverage of considered planners. The bottom-most row provides their overall IPC 2023 learning track score. Our new models are the WL models. Only LAMA-first and Muninn are multi-queue heuristic search planners. The top three single-queue heuristic search planners in each row are indicated by the cell colouring intensity, with the best one in bold. The best planner overall in each row is underlined.

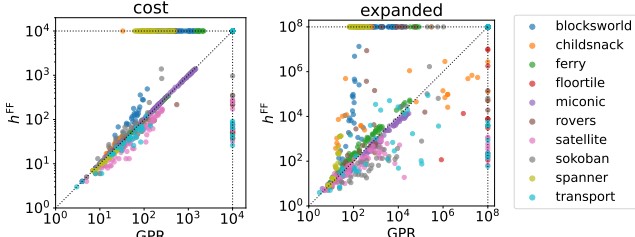

Figure 6: Returned plan cost and number of expanded nodes of $h^{FF}$ and GPR. Problems that were not solved by one planner has their respective metric set to the axis limit. Points on the top left triangle favour GPR while points on the bottom right triangle favour $h^{FF}$.

domain generally has better plan quality and fewer number of expanded nodes. The only exception to this is Sokoban where GPR expands more nodes than $h^{FF}$ but solves more problems because it is faster with its heuristic evaluations. Overall, the domains in which GPR performs worse are domains that require traversing a map which WL features cannot express due to the finite number of WL iterations.

**Are our methods more computationally efficient to train?** To answer this question, we compare the training time of GNNs using ILGs, SVR and GPR. Their mean and 95% confidence interval in seconds are $112.8 \pm 54.3$ (GNN$_{max}$), $142.7 \pm 62.6$ (GNN$_{mean}$), $0.8 \pm 0.5$ (SVR) and $2.6 \pm 1.7$ (GPR). Comparing against the most efficient GNN model per domain, we have that SVR is between 33x (Sokoban) to 421x (Rovers) more efficient and GPR is between 13x (Blocksworld) and 118x (Spanner) more efficient. Note that the GNNs have access to GPUs and they would take even more time to train on a CPU.

| Domain | $h$ error | | | | Expanded | | | |
|---|---|---|---|---|---|---|---|---|
| | E. | M. | H. | all | E. | M. | H. | all |
| blocksworld | **+0.93** | **+0.90** | **+0.94** | **+0.98** | *+0.32* | *+0.22* | *+0.33* | **+0.58** |
| childsnack | **+0.69** | **+0.93** | - | **+0.87** | *+0.59* | *+0.52* | - | *+0.20* |
| ferry | **+0.86** | **+0.98** | **+0.99** | **+1.00** | **+0.86** | **+0.87** | **+0.83** | **+0.93** |
| floortile | - | - | - | - | - | - | - | - |
| miconic | **+0.56** | **+0.67** | **+0.97** | **+0.96** | **+0.55** | **+0.81** | **+0.99** | **+0.99** |
| rovers | **+0.89** | **+0.86** | - | **+0.96** | *+0.26* | *+0.19* | - | **+0.53** |
| satellite | **+0.73** | **+0.95** | - | **+0.96** | *+0.09* | *+0.07* | - | *+0.18* |
| sokoban | *+0.27* | **+0.86** | - | **+0.96** | *+0.26* | **+0.76** | - | **+0.79** |
| spanner | **+0.36** | **+0.53** | **+0.96** | **+0.92** | **+0.43** | **+0.54** | **+0.96** | **+0.92** |
| transport | **+0.83** | - | - | **+0.83** | **+0.37** | - | - | **+0.35** |

Table 2: Pearson's correlation coefficient $\rho$ between standard deviation obtained by GPR against heuristic estimate error and node expansions of initial states from solved problems. Statistically significant coefficients ($p < 0.05$) are highlighted in bold font and italics otherwise. Strongly correlated values ($|\rho| \geq 0.5$) are highlighted in green, medium correlated values ($0.3 \leq |\rho| < 0.5$) in a lighter green, and low correlation ($|\rho| < 0.3$) in gray. Entries for which we solved fewer then 10 problems are omitted. E., M., H. stand for easy, medium and hard difficulties respectively.

**Does kernelising help?** As commonly done in classical machine learning, we combine our WL features with non-linear kernels to obtain new non-linear features that can increase the expressivity of the regression models. Unfortunately, as shown in Tab. 1, this generally results in a decrease in the performance of the learned heuristic: the SVR$_\infty$ model has significantly worse coverage than SVR despite theoretically having more expressive implicit features. The drop in performance can be explained by overfitting to the more expressive features which do not bring any obvious semantic information for planning tasks.

**Do higher order WL features help?** The motivation for using higher-order WL features is similar to using higher-order kernels: to introduce more expressive features that may be correlated with the optimal heuristic. In Tab. 1, we see that the performance of 2-LWL is generally worse on all domains except for Transport. This again can be attributed to poorer generalisation. Furthermore, computing the 2-LWL features are slower to generate than WL features as they take time cubic in the size of the ILGs in the worst case. We also note that attempting to generate 3-LWL features causes out of memory problems during training as the size of features generated is extremely large, on the order of $10^7$ and above.

**Are Bayesian variance estimates meaningful?** One advantage of Bayesian models is that by assuming a prior distribution on the weights of our models, we are able to derive uncertainty bounds on the outputs of the learned posterior model. In Tab. 2, we analyse the Pearson's correlation coefficient between the standard deviation obtained by GPR and (1) the error between output mean and $h^*$, and (2) the number of expanded nodes using the learned heuristic with greedy best first search.

We see that there is a statistically significant strong correlation between the heuristic estimate error and the GPR variance outputs. This is reasonable given that the deriva-

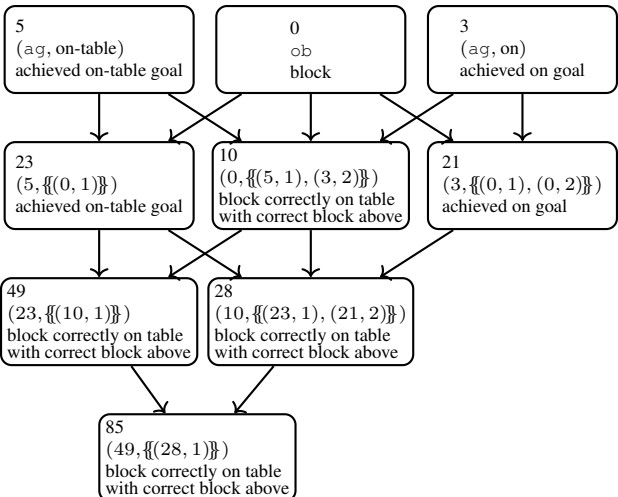

Figure 7: The dependency subgraph of generated WL features on Blocksworld. The first row of each node indicates the index of the feature and also the colour. The second row indicates the initial colour the feature corresponds to or the input to the hash function which generated the colour. The third row describes the semantic meaning of the feature. Edges describe the dependency of the feature on previous features based on the hash function.

tion of the Bayesian model computes the uncertainty on its output prediction. The story is different for the number of expanded nodes during search where for easy problems there is no significant correlation depending on the domain. Interestingly, the correlation is more significant and stronger on harder problems for more domains. Thus, the Bayesian model is able to determine the difficulty of solving a problem within a domain by looking at the predicted standard deviation for $h(s_0)$ but the quality of this prediction will depend on the domain.

**Understanding Learned Models**

Another advantage of WL-GOOSE is that its set of features is explainable, and it is possible to see which features are chosen when using a linear inference model. The models can be understood by analysing the features with the highest corresponding linear weights, and by observing the distribution of such weights. The semantic meaning of the features can be understood by examining the *generation* of WL colours. This can be achieved by representing the observed WL colours as a directed acyclic graph (DAG) where each WL colour is a node and there is a directed edge from $\kappa$ to $\kappa'$ if $\kappa' = \text{hash}(x, M)$ and $x = \kappa$ or $\exists \iota, (\kappa, \iota) \in M$.

We provide an example of how to interpret the learned models by briefly studying the learned GPR model on Blocksworld. In this domain, a total of 8990 features were generated from the training data and Fig. 7 illustrates the DAG representation of feature 85's generation. Consider feature 10 in Fig. 7, it computes the number of blocks that are correctly on the table and also have the correct block above it. We have that $10 = \text{hash}(0, \{\!\{(5, 1), (3, 2)\}\!\})$, mean-

ing that the colour 10 is generated from an object node ($0 = \text{ob}$) which is part of an achieved on-table goal ($5 = (\text{ag}, \text{on-table})$) and achieved on goal ($3 = (\text{ag}, \text{on})$). The corresponding edge label of the node colours indicate the position of the block object in the proposition. Thus, blocks $b$ with colour 10 are in the first and only argument of on-table and the second argument of on. This means that the colour 10 is assigned to blocks correctly on the table and correctly underneath another block.

Moreover, we observed that certain subsets of features were evaluated to the same value on all training states. As a result, the same learned weight value was assigned to each feature in these subsets. This can be seen in Fig. 7 where features 10, 28, 49 and 85 are semantically equivalent. The sum of their weight values is $-1.70$, the second largest in value from subsets of features. Thus, the learned weight rewards states satisfying this condition as blocks correctly on the table do not have to be moved.

Note that, it is possible for features to evaluate to the same values on the training set but have different semantic meanings because the training set is finite. For example, in Blocksworld, a training set may satisfy that a block is correctly on the table if and only if it has the correct block above it. In this case, the count of colours 5 and 10 would be the same on all states despite not being semantically equivalent.

## 6 Conclusion

We introduced WL-GOOSE, a novel approach that combines the efficiency of Statistical Machine Learning (SML) models with features commonly used by Deep Learning (DL) methods. To achieve this, we developed the Instance Learning Graph (ILG), a novel representation of lifted planning tasks and provided a method to generate features for ILGs based on the WL algorithm. Similar to Description Logic features for planning, our generated features are agnostic to the learning target and can be used without the need for backpropagation. Furthermore, unlike DL-based approaches, our models can be trained in a deterministic fashion with minimal parameter tuning. To validate the benefits of WL-GOOSE, we have used two classical SML models, namely support vector regression (SVR) and Gaussian process regression (GPR), to learn domain-specific heuristics and compared them to the state of the art.

The experimental results showed that WL-GOOSE can efficiently and reliably learn domain-specific heuristics from scratch. Compared to GNNs applied to ILGs, our learned heuristics are between 13 to 421 times faster to train and solve up to 28.5% more problems. Our results also showed that both SVR and GPR are the first learned heuristics capable of outperforming $h^{\text{FF}}$ in terms of total coverage. Moreover, our learned heuristics outperform or tie with LAMA on 4 domains and, to our knowledge, this is the best performance of learned heuristics against LAMA. We also showed the theoretical connections between our novel feature generation method with Description Logic features and GNNs. Our future work agenda includes exploring how to best use the uncertainty bounds provided by GPR to improve search, as well as combining stronger satisfying search algorithms to further improve WL-GOOSE.

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
