# OpenReview forum: "Return to Tradition: Learning Reliable Heuristics with Classical Machine Learning"
_icaps-conference.org/ICAPS/2024/Conference — ICAPS 2024_

### Official Review · Reviewer_rKFF · 2024-01-19

**Significance And Importance:** 2
**Soundness:** 4
**Novelty:** 3
**Clarity:** 4
**Overall Evaluation:** 2
**Confidence:** 4

**Weaknesses:**

2: No major or minor weaknesses.

**Contributions Of The Paper:**

This paper introduces a novel system, called WL-GOOSE, that combines statistical machine learning and a new graph representation of lifted planning tasks called Instance Learning Graph (ILG) to learn planning heuristics. ILGs are graphs with coloured nodes and edges that are used to represent planning states. In the paper the authors demonstrate the theoretical connections between ILGs, Description Logic features and GNNs.
The presented approach differs from other recent works by adopting graph kernels instead of  GNNs for learning domain knowledge. These graph kernels are computed using a version of the WL algorithm that processes graphs with coloured nodes and edges executed on the ILG representation of the considered planning state. Finally, the graph kernels are fed to two SML models (i.e. SVR and GPR) to learn the planning domain heuristic.
The experimental results show that WL-GOOSE can learn domain-specific planning heuristics outperforming the coverage of LAMA and hFF in some domains. Furthermore, the training of these models is up to 421 times faster w.r.t. GNNs.

**Ethical Considerations:**

(1) Not Applicable: The paper does not have any ethical considerations to address

**Nomination For Best Paper:**

No

**Questions For Authors:**

1) What are the ranges of objects for the easy, medium and hard problems in each domain?
2) What are the performance on the different classes of problems in terms of coverage?

**Reproducibility:**

3: Authors describe the implementation and domains in sufficient detail.

**Strengths Of The Paper:**

This paper tackles a very active research field and proposes a new way to learn planning heuristics.
The authors introduce a novel representation of lifted planning tasks called ILG. The presented system, called WL-GOOSE, adopts classical SML models trained on graph kernels extracted from ILGs. This process reduces both the hyperparameter selection needed in other state-of-the-art approaches that use GNNs and the training time.
The paper is clearly written and well organized. WL-GOOSE achieves remarkable results in terms of coverage managing to outperform LAMA and hFF in some domains.
The paper is well-organized and clearly written.
The theoretical results proposed are sound.
The examples used in the paper are very helpful for better understanding both the ILG and the theoretical results.

**Weaknesses Of The Paper:**

I think the experimental section of the paper is missing some experimental settings and results. I will address them in the question section.
Although they are well-known, I think a brief background section on the SML models used, similar to the one for Classical Planning, would be useful.

---

> ### Author Rebuttal · Authors · 2024-01-27
>
> # Q1
> Due to space constraints, we have to omit the ranges of objects for the different class of problems in each domain. We will add a reference to the competition page with this information. We will also include a table with this information in the appendix if the paper is accepted
> # Q2
> We will interpret classes of problems as the problem difficulties easy, medium and hard. The table below summarises the coverage over all domains for each difficulty. For GNNs, the values below are the mean and stdev out of 5 repetitions
>
> | class | lama-first | hff | muninn | gnn-max | gnn-mean | svr | svr-inf | 2-lwl | gpr |
> |--|--|--|--|--|--|--|--|--|--|
> | easy | 280 | 275 | 205 | 256.8 ± 3.6 | 250.6 ± 2.6 | 255 | 235 | 235 | 266 |
> | medium | 190 | 112 | 30 | 109.6 ± 7.4 | 105.8 ± 10.3 | 146 | 5 | 92 | 169 |
> | hard | 87 | 43 | 0 | 46.8 ± 4.5 | 41.8 ± 2.6 | 76 | 0 | 7 | 78 |
> | sum | 557 | 430 | 235 | 413.2 ± 10.9 | 398.2 ± 9.8 | 477 | 240 | 334 | 513 |
>
>
> The coverage on the hard difficulty is a good indicator of the overall ranking of a planner with the exception of the GNNs and hFF where the GNN with the mean aggregator has slightly higher coverage than hFF on hard problems but lower coverage overall.
>
> # Background on SML models
> Unfortunately due to space constraints, we have removed the background for the SML models. We have decided to omit them from the background because the main contribution of our work is not the model for learning heuristics, but the feature extraction method for planning problems that can be used for any downstream SML model. We will provide useful references for the models which may provide much better descriptions and intuitions than what we can do with space constraints.

---

### Official Review · Reviewer_HeWG · 2024-01-21

**Significance And Importance:** 1
**Soundness:** 2
**Novelty:** 2
**Clarity:** 2
**Confidence:** 4

**Weaknesses:**

0: Minor weaknesses requiring some work to be addressed for the paper to be accepted.

**Contributions Of The Paper:**

This paper tries to find a new technique to extract knowledge to solve tasks in automated planning. The authors propose to use a tool called WL-GOOSE that uses statistical machine learning and deep learning methods. WL-GOOSE uses a graph representation for lifted planning tasks and generated domain knowledge.  They test their approximation with learning track of the 2023 IPC domains and the results that author show are quite promising.

**Ethical Considerations:**

(4) Good: The paper adequately addresses most, but not all, of the applicable ethical considerations

**Nomination For Best Paper:**

No

**Overall Evaluation:**

-2: (reject)

**Questions For Authors:**

Why do you compare the result with FDSS or GOFAI?

Why table 1 is coverage? The results in learning track are in terms of quality. Lama gest 517.6 FDD 74.1 and SMAC 445.7, and the winner of the competition, GOFAI 508.5.

**Reproducibility:**

3: Authors describe the implementation and domains in sufficient detail.

**Strengths Of The Paper:**

The authors provide a theorical comparison between their approach, GNN, and Description Logic  features in section 3. That comparison shows some differences with previous theories.

Domain learning planners are not a well-studied area and it is interesting to propose new ideas and the experiments are reproducible and well justified.

**Weaknesses Of The Paper:**

I don’t really sure why the authors select Lama-first for comparison. IPC Organizers select two different planners for learning track Fast Downward SMAC 2014 and Progressive Generalized Planner. I recommend comparing with this ones. Especially if we want to frame it in the same framework as the competition, as well as the domains used in it. And the results are in term of coverage not quality like in the competition.

The paper said: “The other planners in the competition were not considered because they do not explicitly learn a heuristic.” I think this is not a good reason to include them in the comparison.

I didn’t know what "Munnin" and there is  no reference in the paper. I think it is important to understand the comparison.

 Small issues:

In the Abstract appear WL algorithm, I suppose should be Weisfeiler-Lehman algorithm, but I don’t sure about that, it is the first time that appear this acronym. In addition, the use of acronyms in the summary is not very common.
Normally when references are used as part of a sentence, only the year should be enclosed in parentheses: “We refer to (Jimenez et al. 2012) for a more, this reference should be We refer to Jimenez et al. (2012)

---

> ### Author Rebuttal · Authors · 2024-01-27
>
> The contribution of the paper is improving ML based heuristics for planning through a new method for extracting features for planning tasks that is agnostic to the downstream ML model. This is done through a theoretically motivated approach. In other words, the goal is not to engineer the fastest planner overall in the literature.
>
> We further note that WL-GOOSE is not an “approximation” and is a complete algorithm as it uses GBFS without operator pruning.
>
> Regarding references to Muninn, it is another entry in the IPC learning track. Please refer to Sec. 4 where it was referenced and discussed in detail and in the setup description in Sec. 5.
> # Q1
>
> The goal of our experimental evaluation is not to compare against scores of a competition that already took place, but instead to compare against similar ML-based methods and the past non-ML benchmarks not surpassed by ML approaches. So far, GBFS+hFF was the non-ML benchmark for learned heuristics and our approach is the first to surpass it. The next benchmark is LAMA (the coverage does not change between LAMA and LAMA-first in Tab. 1). Once ML approaches can beat it, then more recent benchmarks will be considered.
>
> That said, techniques from FDSS and LAMA, and other planning engineering techniques can also be used for WL-GOOSE, e.g., portfolios, multi-queue search, preferred operators, lazy evaluation, novelty heuristics and anytime search. If we also apply these techniques to our planner, we may be able to achieve scores more competitive with SOTA planners but then this hides the base contribution of our work.
>
> We lastly note that the competition winner GOFAI learns a transformation that partially grounds a problem and feeds it into LAMA but the competition result shows that it performs worse than LAMA without the transformation. Thus there is no value in comparing our approach to GOFAI instead of LAMA.
> # Q2
> Please note that the total IPC scores are still presented in the last row of Tab 1. We are happy to provide IPC scores per domain for all planners in an appendix too.
>
> However, the IPC score compresses too much information by summarising two dimensions (plan quality and coverage) into a single value, e.g. the score of a planner X that solves 10 problems with plans 10x worse than the reference plan is the same as the score of a planner Y that solves a single problem with the same plan as the reference one. We decided to show both metrics separately: coverage in Tab 1 and quality in Fig 6 and text.

---

### Official Review · Reviewer_vH2T · 2024-01-22

**Significance And Importance:** 3
**Soundness:** 4
**Novelty:** 3
**Clarity:** 4
**Overall Evaluation:** 1
**Contributions Of The Paper:** 1. WL-GOOSE is a new way of learning …
**Confidence:** 4

**Weaknesses:**

0: Minor weaknesses requiring some work to be addressed for the paper to be accepted.

**Ethical Considerations:**

(5) Excellent: The paper comprehensively addresses all of the applicable ethical considerations

**Nomination For Best Paper:**

No

**Questions For Authors:**

1. What is WL-GOOSE's response time in places with dynamic state space?
2. Any comments on handling WL-GOOSE from overfitting when using more complex features, like non-linear kernels, in complex domains?
3. Please explain more about the concerns that come up when comparing the suggested method's speed to its optimality, especially when a lot of computing power is needed.
4. What does WL-GOOSE do when the domain's world model changes?
5. If the problems were the same level of hardness across all domains, and if the levels of hardness changed for specific items that required more steps to reach the goal state.

- When manipulating robots in ways that require fine or exact movements, the patterns that they need to learn are complicated and subtle. Knowing how WL-GOOSE keeps from overfitting in these situations is very important.

**Reproducibility:**

3: Authors describe the implementation and domains in sufficient detail.

**Strengths Of The Paper:**

1. Combining traditional machine learning with graph models significantly changes methods that mostly use deep learning, making it much more data-efficient.
2. Proven to be efficient with computing, with models running up to 421 times faster than GPU-trained GNNs.
3. Better performance than previous methodologies, with more complete coverage across more planning tasks.
4. Better at generalization than earlier DL-based methods.
5. The study not only explains the method in terms of theory but also backs it up with data from the International Planning Competition.

**Weaknesses Of The Paper:**

1. This paper shows better generalization than earlier ones, but it doesn't entirely look into how it can be used for many different planning tasks. There are concerns about how flexible the method is in difficult real-life situations because of this limitation.
2. There are problems with overfitting when non-linear kernels and higher-order WL features are added. This affects the performance. Because it is too good at fitting, the model might miss important dynamics in planning tasks with complex patterns, like manipulating robots in complex ways or making strategic decisions in games.
3. This model's experimental results show it is efficient, but there isn't enough in-depth research on the trade-off between how fast it works and how well it solves problems. This trade-off is significant for tasks requiring much computing power, like large-scale and complex domains.
4. Only certain areas from the International Planning Competition are used for the trials. Many test cases are needed to understand how robust the method is entirely.
5. The model's experimental setup doesn't mention the problem's difficulty; even though it focuses on generalization in terms of objects and domains, the lack of information related to the difficulty of the problem instances doesn't assure the correct interpretations of the results in terms of performance.
6. Missing citations relevant to the publication, such as https://dspace.cuni.cz/handle/20.500.11956/180047 and others.

---

> ### Author Rebuttal · Authors · 2024-01-27
>
> # W4
> The goal of the paper is to evaluate a new ML method for classical planning, and this is what our selected benchmark suite (the Learning Track of the IPC) is designed to evaluate. The benchmarks have been selected by the IPC organisers to consist of a diverse set of classical planning tasks.
>
> # W5/Q5
> The organisers of the competition selected problems based on hardness and classify the problems into easy, medium and hard (see beginning of Sec 5). Please, see the table on our answer to Q2 rKFF for the results of our experiment for each level.
>
> Note that all planners are operating on the same benchmark and experimental setup so there is no incorrectness in interpreting the results. If in a domain the problems are harder (easier) than in other domains, all planners will be penalised (take advantage). In other words, relative performance is more important than absolute performance.
>
> # Q1/4
> No planner considered in our paper (or in the IPC learning track) can handle changes in the domain/state space. All of them do off-line planning. That said, WL-GOOSE using ILG and other graph representations can be used to learn a domain-independent heuristic that can be used on any PDDL domain, or a heuristic that can adapt to changes in the model.
>
> Moreover, it is possible to extend our model to the online planning setting. This is because our models are cheap to retrain, which is a benefit for online planning over dynamic state spaces or world models, and it is also possible to apply RL since our feature generation of planning states is agnostic to the choice of model.
>
> # Q2
> The generated features of our approach are agnostic to the choice of learning model, in contrast to neural network architectures which use gradient descent for generating features. Thus we can use more models that are less prone to overfitting such as linear methods for more complex domains. In other words, overfitting is not a problem of our architecture but a choice of the downstream ML model.
>
> # Q3
> This tradeoff is captured in the IPC score shown in the last row of table 1. A planner receives a score C*/C for each problem solved, where C* is the cost of a reference plan, and C is the cost of the returned plan. A score of 0 is given to unsolved problems. Fig. 6 also shows the plan cost, i.e., quality of solution, between hFF and WL-G. Lastly, it is possible to allow for a trade-off between speed and optimality by using anytime search such as in the LAMA planner.

---

### Meta-Review · Area_Chair_S4ms · 2024-02-06

**Recommendation:** Accept (Oral)
**Confidence:** 4

**Metareview:**

This paper proposes an approach to learning domain-specific heuristic functions for classical planning which uses a graph representation and statistical learning methods.

Reviewer opinions were divided on this paper. Unfortunately, there was little discussion after the author rebuttal phase.

As metareviewer, I tend to agree with the assessment of paper strengths cited by the reviewers. Specifically, the paper proposes a novel graph representation (ILG), theoretically analyzes the capabilities of this model, and shows impressive experimental results compared to recent, domain-specific learning systems.

 The reviewer with the lowest score seems to base the low score on perceived weaknesses with the experimental evaluation, but I believe  the authors rebuttal adequately addresses these concerns, and I believe the experimental evaluation is convincing with respect to the goals of this paper as explained by the authors.

**Ethical Considerations:**

(1) Not Applicable: The paper does not have any ethical considerations to address